# Future Image Synthesis for Diabetic Retinopathy Based on the Lesion Occurrence Probability

**Sangil Ahn** [1], **Quang T.M. Pham** [1], **Jitae Shin** [1,*] **and Su Jeong Song** [2,3*]

1 College of Information and Communication Engineering, Sungkyunkwan University, Suwon 16419, Korea; il2s@skku.edu (S.A.); quangpham@skku.edu (Q.T.M.P.)
2 Biomedical Institute for Convergence (BICS), Sungkyunkwan University, Suwon 16419, Korea
3 Department of Ophthalmology, Kangbuk Samsung Hospital, Sungkyunkwan University School of Medicine, Seoul 03181, Korea
* Correspondence: jtshin@skku.edu (J.S.); ssjeye@skku.edu (S.J.S.)

**Abstract:** Diabetic Retinopathy (DR) is one of the major causes of blindness. If the lesions observed in DR occur in the central part of the fundus, it can cause severe vision loss, and we call this symptom Diabetic Macular Edema (DME). All patients with DR potentially have DME since DME can occur in every stage of DR. While synthesizing future fundus images, the task of predicting the progression of the disease state is very challenging since we need a lot of longitudinal data over a long period of time. Even if the longitudinal data are collected, there is a pixel-level difference between the current fundus image and the target future image. It is difficult to train a model based on deep learning for synthesizing future fundus images that considers the lesion change. In this paper, we synthesize future fundus images by considering the progression of the disease with a two-step training approach to overcome these problems. In the first step, we concentrate on synthesizing a realistic fundus image using only a lesion segmentation mask and vessel segmentation mask from a large dataset for a fundus generator. In the second step, we train a lesion probability predictor to create a probability map that contains the occurrence probability information of the lesion. Finally, based on the probability map and current vessel, the pre-trained fundus generator synthesizes a predicted future fundus image. We visually demonstrate not only the capacity of the fundus generator that can control the pathological information but also the prediction of the disease progression on fundus images generated by our framework. Our framework achieves an F1-score of 0.74 for predicting DR severity and 0.91 for predicting DME occurrence. We demonstrate that our framework has a meaningful capability by comparing the scores of each class of DR severity, which are obtained by passing the predicted future image and real future image through an evaluation model.

**Keywords:** future diabetic retinopathy image synthesis; prediction occurrence probability; generative adversarial network





## 1. Introduction

Diabetic Retinopathy (DR) is a disease in which micro-blood vessels in the retina are damaged, and related blood vessel damage is one of the most fatal complications of diabetes. DR is a major cause of blindness worldwide. Neovascularization, which can be observed at all severities of DR, results in poor vascular wall structures, which can cause internal materials to leak out of the blood vessels or burst blood vessels, causing inflammation and edema and obstructing vision. In particular, bleeding or edema in the central part of the retina (macula), where the visual cells are concentrated and projected images are in focus, can lead to severe vision loss and blindness. This symptom is called Diabetic Macular Edema (DME). All patients with this DR disease potentially have a DME symptom. Therefore, predicting the occurrence of symptoms of DME in advance is important in predicting future DR severity from current information to preserve the vision of patients.

Nowadays, deep learning is used in various medical fields as a technique for analyzing medical images. Rachmadi et al. [1] predicted the evolution of White Matter Hyperintensities in small vessel disease based on a Generative Adversarial Network (GAN) model and an irregularity map. Xia et al. [2] synthesized brain images that vary over time by learning a joint distribution of the patient age and brain morphology to predict the future state of the brain. Schlegl et al. [3] proposed fast unsupervised anomaly detection with a GAN, which is able to detect the abnormal parts of an image using only healthy OCT images. Baumgartner et al. [4] used 2D ultrasound data for detecting 13 fetal standard views with convolutional neural networks. Guan et al. [5] generated synthetic mammography images to use as a dataset for detecting abnormal volumes in the breast. In particular, Deep Convolutional Neural Networks (DCNNs) are mainly used to check the condition of fundus disease. Khojasteh et al. [6] compared several deep learning methods to maximize the sensitivity and specificity for detecting exudates. To deal with multi-class classification to automatically classify the severity of DR, Wang et al. [7] employed the deep learning-based Convolutional Neural Network (CNN) for DR severity classification, and Zhao et al. [8] introduced a model that combines an attention model for feature extraction and a bilinear model for classification. For improving the capacity of both the disease severity classification and semantic segmentation task, Zhou et al. [9] proposed a collaborative learning method that uses semi-supervised learning with an attention mechanism. To diagnose these symptoms, Ren et al. [10] proposed a screening system that classifies the severity of DME by considering the representative features and vector quantization. Syed et al. [11] used knowledge of exudates and macula for the grading of DME severity.

There are relatively few studies that diagnose the fundus state of patients who have a DR or DME lesion. There are further significant shortages of studies that predict the future state of the fundus after a certain period of time. To predict the progression of DR, Arcadu et al. [12] proposed a deep-learning algorithm to predict the risk of worsening DR in individual patients. However, to the best of our knowledge, there is no work for synthesized fundus images for retinal disease progression. There are some problems, such that a large quantity of data is needed to generate images, and longitudinal data from the same patient is needed to predict the future state. It is also hard to correct paired fundus images from the same patient over time. Moreover, even though we collect the longitudinal images from the same patients, the images may have different lesion locations because the angle of the eye and equipment is different for each test.

To overcome the challenges and analyze the progression of DR disease, we propose a future prediction framework to synthesize the future fundus image for visualizing the future state of the fundus. Our framework has been trained with two training steps: (1) Fundus generator training, and (2) lesion probability predictor training. For the first step, we first train a fundus generator to synthesize a realistic fundus image that requires a lesion segmentation mask and a vessel segmentation mask with an adversarial method. Since the fundus generator produces realistic fundus images with only vessel and lesion information, various DR datasets can be used as data needed for learning, if we are able to extract vessel and lesion information from the fundus image. Therefore, we do not need longitudinal data for training in the first step. Then, the trained generator can synthesize a realistic fundus image no matter what vessel mask and segmentation mask are used as input values. By using this pre-trained generator, we again train a lesion probability predictor with the adversarial method. In the training step of the lesion probability predictor, we give a current fundus image and a vector representing the interval between the future and current image to the lesion probability predictor. Then, the predictor predicts a probability map that has the same size as the given current image and indicates where the lesion will occur. It can be used as a future lesion segmentation mask. During the training of the lesion probability predictor, we need to compare the difference between the real future lesion segmentation mask and the predicted probability map. However, even though we performed pre-processing with a regularization method to reduce the difference between the current fundus image and future fundus image on the pixel level, lesion-coordinate

differences between the two images for the longitudinal data can still occur. To solve this problem, we introduce a regularization loss function to give a penalty of this pixel level distinction. By calculating the difference between the real current vessel coordinate and future vessel coordinate, the larger the calculated difference value, the smaller the effect of the regularization loss on the total loss. This regularization loss can help the lesion probability predictor accurately generate the probability map. Finally, the probability map and the current vessel mask are given to the pre-trained fundus generator, then we can synthesize the predicted future fundus image by passing through the pre-trained fundus generator. Overall, our contributions to this research are described as follows:

- We propose a future prediction framework for future fundus image synthesis of patients with DR to observe a progression of the disease by preserving the vessel identity of patients based on an adversarial learning mechanism. The framework, which consists of a fundus generator and a lesion probability predictor, effectively addresses the issue of longitudinal patient data shortages for future image visualization by separating the training sequence.
- We combine the reconstruction loss and an adversarial loss to improve the performance of fundus image synthesis to effectively control the pathological information by using the lesion and vessel mask in fundus generator training. Furthermore, in the lesion probability predictor training step, we introduce a regularization loss function to accurately predict the lesion occurrence probability by calculating the structural similarity between the current vessel and target future vessel at the pixel level.

## 2. Methods

Our proposal aims to synthesize the future fundus images based on the current state of information to observe the progression of the disease with a deep learning method. As shown in Figure 1, we introduce a prediction framework for the synthesis of a future fundus image with two training steps. In the first step, the fundus generator, employing a mechanism based on a Conditional Generative Adversarial Network (cGAN), synthesizes a realistic fundus image by using a lesion segmentation mask and vessel segmentation mask. In the second step, the lesion probability predictor takes the current fundus image and interval vector, which is the difference between two time periods, and creates a probability map that can be used as the future lesion mask. The probability map shows a position where the symptom of the disease will occur after the interval time period. Then, the generator, which is pre-trained in the first step, takes the probability map and a current vessel mask to generate a predicted future fundus image.

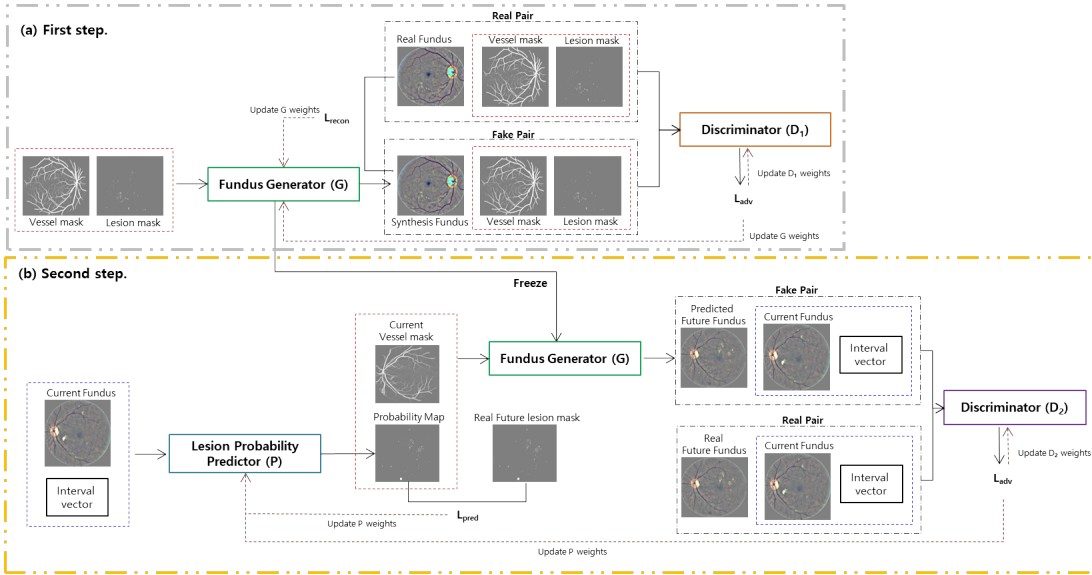

**Figure 1.** The proposed future-prediction framework by separating the two training steps for efficient learning: Fundus generator training and lesion probability predictor training.

## 2.1. First Step: Fundus Generator Training

The purpose of the first step is to have a capacity for synthesizing a realistic fundus image. Since this is an image generation task, we employ an adversarial learning mechanism that is widely used for image synthesis based on given inputs. We build two models: A fundus generator ($G$), and a discriminator ($D_1$). The first step in Figure 1 shows the fundus generator training process. The generator $G$ takes the two inputs of the lesion segmentation mask ($l$) and vessel segmentation mask ($v$). Micro aneurysms, hemorrhage, soft exudates, and hard exudates are typical lesions observed in DR patients. The lesion mask used in this paper is a one-channel image representing the four representative lesions. The masks $l$ and $v$ are passed through $G$, then $G$ generates a synthesized fundus image, $S_f$, which resembles the real fundus image, $O_f$, based on the lesion and vessel information. Simultaneously, the discriminator $D_1$ tries to learn the distribution of both $O_f$ and $S_f$ to distinguish them.

Even though we employ an adversarial learning mechanism that normally uses only the adversarial loss function for training, $G$ needs to be trained to have a capacity that accurately synthesizes the identity of the lesion and vessel in the same position on $S_f$ compared to $O_f$. This is similar to the process that reconstructs the real image by using the elements extracted from the real image. To further ensure that the identity of lesion and vessel of $S_f$ will be located in the same position as $O_f$, we combine the adversarial loss ($\mathcal{L}_{adv}$) and the reconstruction loss ($\mathcal{L}_{recon}$) to have a pixel-space benefit and to improve the performance of image synthesis. Furthermore, to classify the real fundus image from the synthesized fundus image, and to determine whether the synthesized fundus image has the lesion in the same location as $O_f$, we use $O_f$, $l$, and the $v$ as a real pair set, and $S_f$, $l$, and $v$ as the fake pair set for training $D_1$. To train $G$ and $D_1$, the total loss function $\mathcal{L}_{step1}$ in the first step is defined as:

$$\mathcal{L}_{step1} = \mathcal{L}_{adv} + \mathcal{L}_{recon} \tag{1}$$

$$\mathcal{L}_{adv} = \mathbb{E}_{l,v,O_f \sim p_{data}}[log(D_1(O_f,l,v))] + \mathbb{E}_{l,v \sim p_{data}}[log(1 - D_1(S_f,l,v))] \tag{2}$$

$$\mathcal{L}_{recon} = \mathbb{E}_{l,v,O_f \sim p_{data}} \mathcal{L}_1(O_f,S_f). \tag{3}$$

## 2.2. Second Step: Lesion Probability Predictor Training

Our aim is to predict future retina states and to synthesize a future fundus image, which can be used to diagnose and understand the progression of the disease. The second step in Figure 1 shows an overview of the lesion probability predictor training. We employ three networks: A lesion probability predictor ($P$), the generator ($G$) that is trained in the first step, and a discriminator ($D_2$).

To generate future images, we use the time interval information between a given current fundus image and a target future fundus image, calculated by day. After extracting all the interval information (e.g., the longitudinal data) in advance, we divide it by the maximum interval value and multiply it by 100 to make the value of interval information as a non-integer number among [0, 100]. At this time, we only use natural numbers. Then, using an ordinal encoding method, we make a 100-dim ordinal interval vector. For example, if the interval value is 30, the first 30 elements are set as 1, and the others are 0 in the interval vector.

Given a current fundus image ($C_f$) and an interval vector ($i$) corresponding to $P$, we predict a probability map, $p_m$, which involves the occurrence probability of future lesions that will occur on the retina after the interval. Since $p_m$ can play a role as a future lesion segmentation mask, the pre-trained G can synthesize a predicted future fundus image ($P_f$) based on the probability information by taking $p_m$ and a current vessel mask $c_v$ as inputs. This training mechanism allows to preserve the identity of the current vessel information. Then $D_2$ takes $P_f$, $C_f$, and $i$ to distinguish them as a fake sample, and takes the real future fundus image ($R_f$), $C_f$, and $i$ to distinguish them as a real sample. During the training phase, $D_2$ learns two tasks: (1) Whether the predicted future fundus image is real or not and

(2) whether the relationship between the current image, future image, and time information is well considered. For training $P$ and $D_2$ without $G$, the adversarial loss $\mathcal{L}_{adv}$ is defined as:

$$\mathcal{L}_{adv} = \mathbb{E}_{R_f,C_f,i\sim p_{data}}[logD_2(R_f,C_f,i)] + \mathbb{E}_{C_f,i\sim p_{data}}[log(1 - D_2(P_f,C_f,i))]. \tag{4}$$

To accurately predict the position of the lesion that will occur to create a probability map, the network $P$ needs to be trained by considering a pixel-level comparison with the target future lesion mask $f_m$. Even though we collect the longitudinal fundus images of the same size from the same patients, the images may have slightly different locations for the lesions because of various reasons, for example the angle between the equipment and patient can be changed whenever taking a fundus image and the equipment can change over time. For those reasons, even if we apply image pre-processing, such as registration to reduce the pixel-level difference between the current fundus image and the real future fundus image, we will inevitably obtain a difference. Therefore, the lesion masks extracted both from the current fundus image and the real future fundus image have different positions of lesions. These differences may lead to the calculation of different positions between the probability map and target future lesion mask at a pixel level. This can interfere with the training of $P$ to predict the progression from the current state. Thus, if the coordinate between the current vessel mask $c_v$ and the target future vessel mask $f_v$ is different at the pixel level, a penalty that is as much as the difference should be given to the loss function. The Mean Square Error (MSE) is a function that can be used to calculate the similarity of images, and the pixel-wise distance (difference) between the two images is averaged to determine the similarity. Thus, the MSE has a large value if the similarity of the two compared images is small. For enhancing the prediction capacity of $P$, we introduce a regularization loss function using the MSE that allows it to give a penalty, which is the difference between $c_v$ and $f_v$ on the pixel level when we train $P$, as:

$$\mathcal{L}_{pred} = \mathbb{E}_{C-f,i,f_m,c_v,f_v\sim,p_{data}} \mathcal{L}_{dice}(p_m,f_m) * \frac{1}{1 + \frac{1}{mn}\sum_{i=0}^{m-1}\sum_{j=0}^{n-1}[c_v(i,j) - f_v(i,j)]^2} \tag{5}$$

$$\frac{1}{1 + \frac{1}{mn}\sum_{i=0}^{m-1}\sum_{j=0}^{n-1}[c_v(i,j) - f_v(i,j)]^2} \triangleq \frac{1}{1 + MSE} \tag{6}$$

where $\mathcal{L}_{dice}$ is a Dice coefficient loss [13] that is widely used as a metric to calculate the similarity between two images. The Dice loss is used to calculate the difference between $p_m$ and $f_m$, and $m$ and $n$ are the respective height and width of $c_v$. The regularization term (in Equation (6)) captures the fact that if the difference of pixel values between the current vessel mask and the target future vessel mask is large, the regularization term should be small. Thus, the regularization loss function helps to reduce calculation errors from abnormal and normal positions at the pixel level. We combine $\mathcal{L}_{adv}$ and $\mathcal{L}_{pred}$ for the second step of the training loss function. The overall training loss is defined as:

$$\mathcal{L}_{step2} = \mathcal{L}_{adv} + \mathcal{L}_{pred}. \tag{7}$$

## 3. Experiments and Result

### 3.1. Datasets

We collect three datasets: The ISBI2018 [14] dataset, EyePACS [15] dataset, and the Kangbuk Samsung Hospital dataset.

**ISBI2018:** The ISBI2018 dataset consists of three kinds of data for the sub-challenge, lesion segmentation, disease grading, the optic disc, and fovea detection. The disease grading purpose in the challenge is to directly compare the methods developed for automatic image grading for DR and DME. We use only disease grading data. The medical experts graded the full set of 516 images with a variety of pathological conditions of DR and DME with 516 images, which consist of 5 classes for the DR severity scale and 3 classes for the risk of DME.

**EyePACS:** These sources provide a large DR dataset of high-resolution retina images taken under a variety of imaging conditions. A clinician divides the five classes of DR severity (No DR, Mild, Moderate, Severe, and Proliferative DR). The EyePACS data contains DR images with artifacts that we may encounter as noise when using the images. There are about 35,000 images, but we choose 30,000 images by excluding the images that have a lot of artifacts.

**Kangbuk Samsung Hospital dataset:** The Kangbuk Samsung Hospital dataset is a longitudinal dataset with a collection of the fundus images of DR patients observed over 10 years, which serves as a disease progression dataset. Several fundus images were obtained from each patient at different times, and the first of the images obtained is considered the current fundus, and then the others obtained are referred to as a future fundus on the forthcoming longitudinal date. The dataset contains 4231 of the fundus images from 1184 subjects who have DR and in addition, 224 of the DME fundus images are included from the subjects who have the disease of DR. For the first step, we selected 200 fundus images for validation and the rest for training. Otherwise, we would need to build pair data, which consists of the current fundus image and future image including the DME disease, for training in the second step. We therefore built 724 pairs of data for training and 100 pairs of data for validation in the second step.

### 3.2. Pre-Processing

Since the fundus images from different datasets have various illuminations, we must clarify the fundus images so as to enable the model to learn the feature more effectively, we resized all DR data to a $3 \times 512 \times 512$ size and then applied a Gaussian blur to bring out distinctive features in the images. In the Gaussian blur operation, the image is applied with a Gaussian filter, which is a low-pass filter, to remove the high-frequency components. Generally, a fundus image is a color image, but in this paper, we call the image applied with a Gaussian filter, a fundus image. To train the fundus generator in the first step, we require a lesion mask and a vessel mask. Using the Kangbuk Samsung Hospital data, an expert made lesion annotations by manually segmenting the micro aneurysms, hemorrhage, soft exudates, and hard exudates. Those conditions and symptoms are important factors in DR. Then, we built a segmentation model to extract the important factors and apply the segmentation model to extract the lesion mask from other data for creating a pseudo lesion mask. We also employed the pre-trained vessel segmentation model [16] for the role of a vessel extractor to obtain the vessel mask from all datasets. Using DRIVE data [17], the vessel segmentation model [16] is trained based on an adversarial learning method for vessel segmentation from the fundus image.

### 3.3. Implementation

We trained $G$ and $P$ with reconstruction loss and regularization loss to improve the respective generation and prediction performance by employing the ResNet-based conditional. In the first step, we used the ISBI2018 disease grading data, EyePACS data, and the Kangbuk Samsung Hospital data for training the $G$ and the $D_1$. In the second step, we only used the Kangbuk Samsung Hospital data for training $P$ and $D_2$, without $G$, by freezing all parameters of the $G$. We set the batch size to 10 and used the Adam optimizer with $\beta_1 = 0.5$, $\beta_2 = 0.999$ and the learning rates for $P$, $G$, $D_1$, and $D_2$ were set to $10^{-4}$, while those of the lesion segmentation model were set to the default for $\beta_1$, $\beta_2$, with only Dice loss function. In the first step and the second step, we set 100 epochs for training $G$ and $D_1$ using approximately 32,000 images and set 2000 epochs for training $P$ and $D_2$ with 724 pair images respectively. Since we used a lot of data to train $G$ and $D_1$, we were able to obtain the $G$ and $D_1$ even if we set up a relatively small number of epochs. Our framework aims to synthesize future fundus images to predict future lesion states. The important thing is to know the stage of a lesion if it happens to be found in the future image. Therefore, we excluded normal cases in future fundus images when we set the current-future pair of fundus images using the Kangbuk Samsung Hospital dataset. Additionally, the mild case

and moderate case had similar features and did not have meaningful clinically significant changes. Consequently, we demarcate significant clinical differences by categorizing the DR severity as mild, severe, or Proliferative Diabetic Retinopathy (PDR).

### 3.4. Qualitative Results

**Image Synthesis of the Fundus Generator:** Once the fundus generator is trained, we can synthesize not only the realistic fundus images, which can be coarsely predicted from the corresponding grading images, but also control the pathological information by switching the lesion mask. We randomly selected 2500 images from each different DR grading for validation and the rest for training by using three datasets. As we can see in Figure 2, the results of the fundus generator show the similarity between the synthesized fundus images and real fundus images for a given input value with lesion information and vessel structure. We observe that the fundus generator can synthesize fundus images that reflect the lesion information and vessel structure within the corresponding two masks at the same position. Moreover, in order to demonstrate the capability of controlling the pathological information arbitrarily, we synthesized the DR fundus images with different DR lesion masks. As shown in Figure 3, based on the fundus images which are real fundus images in the red box, the other synthesized fundus images contain different pathological information from different lesion masks by preserving vessel identity. Therefore, the generator can synthesize future fundus images by preserving the current vessel information based on the predicted probability mask in our framework.

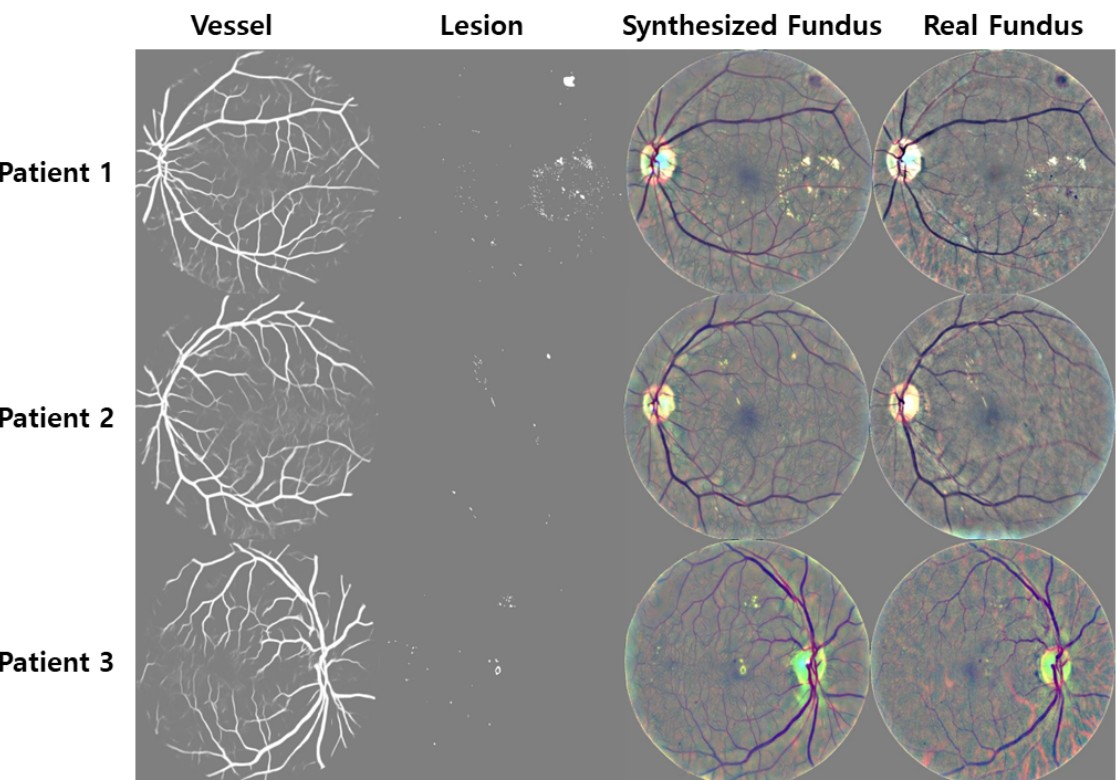

**Figure 2.** Images showing the synthesized samples of the fundus generator based on the vessel mask and lesion mask.

**Predicted Future Fundus Images:** We show several examples of the predicted future fundus images from 4 patients and visualize the differences between the synthesized fundus images and real fundus images for the qualitative comparison in Figure 4. From the results, we can find that the lesions are not only located in the same position based on the vessel structure but also predicted the changing pattern over time. Furthermore, we also have made almost similar predictions for the patient that has no significant variation in the lesions, even though we only employ a binary segmentation mask as the lesion mask.

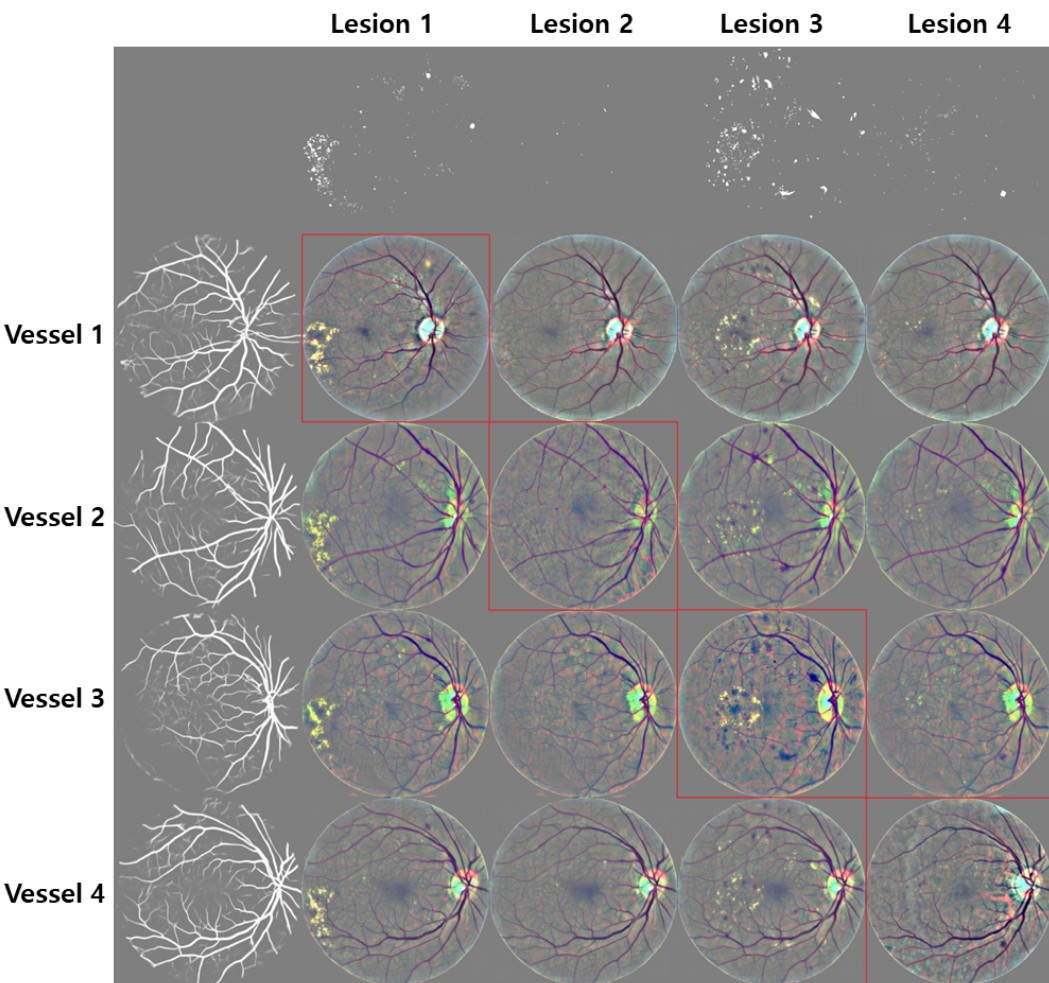

**Figure 3.** A qualitative comparison of the fundus generator for fundus image synthesis. The fundus generator can synthesize the fundus image by changing the lesion mask with a fixed vessel mask compared to the real fundus images, which are in the red box.

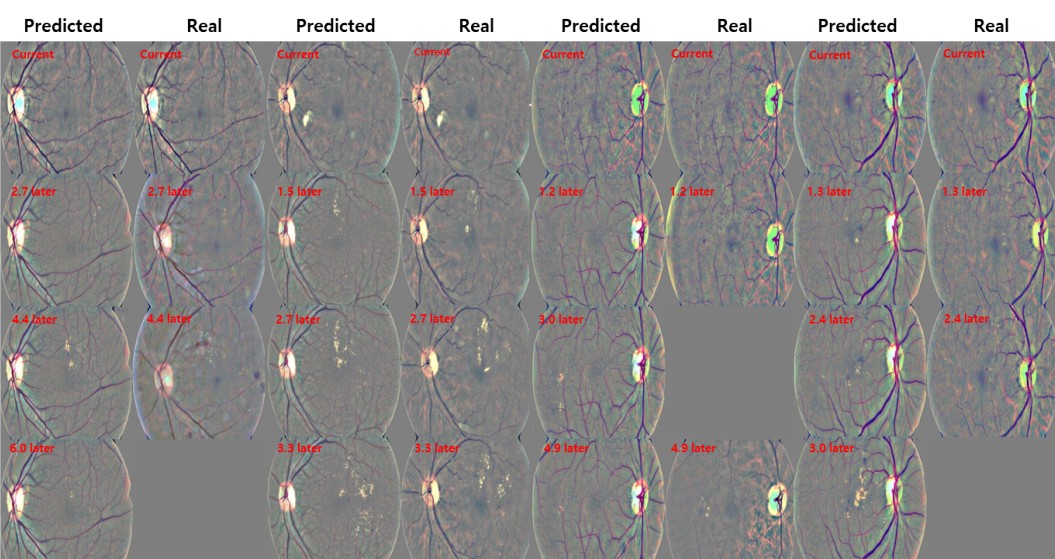

**Figure 4.** The sample results of predicted future fundus image produced with the regularization loss function. We can properly predict the progressions of different diseases from each patient over time and confirm them by synthesizing the future fundus image. The text label (e.g., 2.7 later) means a predicted future image after 2.7 years from the current input image.

### 3.5. Quantitative Results

The important concern in our framework is to evaluate that how well the framework generates predicted future fundus images to observe the progression of the disease over time. We compared four methods: cGAN [18], Tub-sGAN [19], and *Ours* (n), and *Ours* (r). We trained cGAN [18] and Tub-sGAN [19], *Ours* (n), and *Ours* (r) to adjust the models for our purpose using 724 pairs of data for training and 100 of pairs data for validation from the Kangbuk Samsung Hospital dataset. To synthesize future fundus images, we train cGAN [18] with the current fundus image and interval value to directly obtain future fundus images. Tub-sGAN [19], which needs a vessel mask and random noise $z$ to synthesize realistic fundus images, was trained based on a vessel mask and a lesion mask and an interval vector to generate a future fundus image. *Ours* (n), *Ours* (r) means that we use normal Dice pixel-level loss and the regularization loss in the second step respectively.

For assessing each method capacity that predict the DR severity and DME occurrence from the predicted future fundus image, we built two proxy evaluation models with the well-known classification models of ResNet-50 [20], EfficientNet-b0 [21], and Se-ResNet-50 [22]. To build two proxy evaluation models, we used 724 future fundus images for training and 100 future fundus images for validation, with image-level expert grades from the Kangbuk Samsung Hospital dataset. Each image has DR and DME grade information.

**DR Severity Evaluation:** As illustrated in Table 1, the F1-scores are employed for the evaluation, and SE-Resnet-50 [22] has the best performance. Thus, we choose SE-Resnet-50 [22] as the DR severity evaluation model for evaluating the predicted future fundus image. Table 2 presents a comparison of the image-level DR severity detection accuracy, specificity, sensitivity, and F1 score. Most importantly, our framework, trained with two steps, has a higher accuracy than all other approaches. Furthermore, based on the comparison results of *Ours* (n) and *Ours* (r), we were able to confirm that the regularization loss helps to generate accurate images. In addition, to demonstrate that our proposal has a meaningful capacity for prediction, the predicted future images and real future images are passed through the SE-ResNet-50 [22] evaluation model to obtain severity prediction results. Then, we compared the ground truth of severity and the prediction results of each class. Table 3 shows that the performance results of the predicted future fundus images are similar to the result from real future fundus images. Therefore, we can indirectly confirm that the images predicted by our proposal have DR severity information similar to that of the real future fundus images.

**Table 1.** Performance of the models for Diabetic Retinopathy (DR) severity.

| Model | F1 Score |
|---|---|
| ResNet-50 [20] | 0.68 |
| EfficientNet-B0 [21] | 0.72 |
| Se-ResNet-50 [22] | 0.79 |

**Table 2.** Evaluation of the DR severity prediction. Conditional Generative Adversarial Network: cGAN.

| Method | Accuracy | Specificity | Sensitivity | F1 Score |
|---|---|---|---|---|
| cGAN [18] | 0.64 | 0.73 | 0.46 | 0.47 |
| Tub-sGAN [19] | 0.78 | 0.84 | 0.69 | 0.62 |
| *Ours* (n) | 0.80 | 0.85 | 0.70 | 0.69 |
| *Ours* (r) | 0.83 | 0.87 | 0.74 | 0.74 |

**Table 3.** Comparison DR severity classification results of each class using the F1-score.

| Data | MILD | SEVERE | PDR | AVG |
|---|---|---|---|---|
| Predicted Future Fundus | 0.80 | 0.71 | 0.70 | 0.74 |
| Real Future Fundus | 0.82 | 0.78 | 0.78 | 0.79 |

**DME Occurrence Evaluation:** We also conducted experiments on whether our method correctly predicted the occurrence of DME disease, which can cause severe vision degradation or blindness, by using the predicted future fundus images from each model. The evaluation model was conducted in the same way as the one used to select the DR Severity evaluation model, and the best-performing ResNet-50 [20] model was chosen from Table 4. In this case, it was divided into DME occurrence and DME non-occurrence based on the presence of disease around the macular region. In terms of DME occurrence, as shown in Table 5, the *Ours* (r) method with the regularization term obtained the best score.

**Table 4.** DME occurrence performance evaluation.

| Model | F1 Score |
|---|---|
| ResNet-50 [20] | 0.94 |
| EfficientNet-B0 [21] | 0.89 |
| Se-ResNet-50 [22] | 0.92 |

**Table 5.** Evaluation for DME occurrence prediction.

| Method | Accuracy | Specificity | Sensitivity | F1 Score |
|---|---|---|---|---|
| cGAN [18] | 0.78 | 0.74 | 0.82 | 0.77 |
| Tub-sGAN [19] | 0.82 | 0.78 | 0.85 | 0.82 |
| *Ours* (r) | 0.94 | 0.95 | 0.92 | 0.91 |

## 4. Discussion and Limitations

Although our method can achieve compelling results in many cases, we did not uniformly generate a fundus image with the information we wanted. Several typical failure cases are shown in Figure 5a. On the fundus generation task based on the lesion mask and vessel mask, the synthesized fundus image has different lesion information compared to the real fundus image. For example, the real has Micro Aneurysms symptoms in the center of the image, but the fundus generator synthesized Hard Exudates symptoms at the same position because the fundus generator was trained on only a 1-channel combined lesion mask that involved four sets of lesion information. We also observed poor performance in the results of predicting symptoms, when there was not enough information of the lesion in the current image. For example, the future image was not changed compared to the current image, as seen in Figure 5b. In addition, the surgical marks were not synthesized.

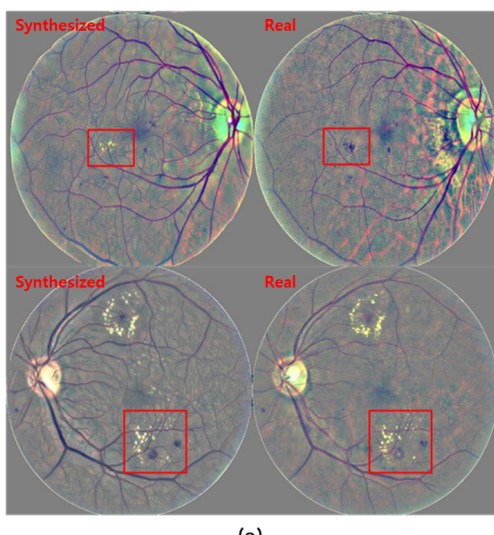
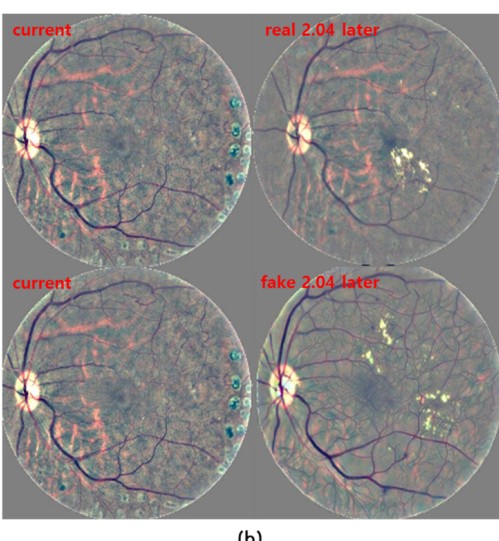

(a)                                                              (b)

**Figure 5.** In (**a**), the synthesized lesion is shown by comparing the lesions in the red box, unlike the real lesion, and in (**b**), the future image is synthesized by predicting differently from the progression of the real lesion.

For removing various illumination features and effectively learning the symptoms of DR disease change, we synthesized a fundus image which was modified by the Gaussian blur operation, not a color fundus image. This method may have been able to efficiently learn our model to generate fundus images. However, it can lead to confusion when experts judge the state of the lesion by looking at the synthesized images. Therefore, we showed the result of quantitative evaluation using evaluation models. For future research, we will study the synthesis of color fundus images to correctly predict each lesion progression with four lesion masks and will obtain evaluation results from a clinical expert.

## 5. Conclusions

We proposed a framework with a two-step training approach: A fundus generation training step and a lesion probability prediction training step. By using several datasets for training the fundus generator, we synthesized the fundus image, which is reflected in the lesion and vessel information. We could train the fundus generator more easily because we could use several simple DR fundus datasets and not longitudinal data. Then, we trained the lesion probability predictor which makes a probability map that has lesion occurrence probability information from the current fundus image given an interval vector. The pre-trained generator takes the probability map and current vessel mask and synthesizes predicted future fundus images. Of course, a large amount of data is also needed to learn a model that predicts the probability of lesions appearing, but a relatively small amount of data may be needed to learn a model that produces images simultaneously with predictions. Using our proposal, we could visually observe the progression of the disease over time. In addition, we achieved F1-scores of 0.74 and 0.91, which are the best prediction performances of DR severity and DME occurrence, respectively, compared to related state-of-the-art methods. We also demonstrated the meaning of our proposal by obtaining a similar score in comparison with real data for each class of DR severity.

**Author Contributions:** Conceptualization, S.A., S.J.S. and J.S.; methodology, S.A. and J.S.; formal analysis, S.A., Q.T.M.P. and J.S.; software, validation, Writing—Original draft preparation, S.A.; Writing—Review and editing, S.A., Q.T.M.P. and J.S.; supervision, project administration, and funding acquisition, J.S. and S.J.S. All authors have read and agreed to the published version of the manuscript.

**Funding:** This work was partly supported by a National Research Foundation of Korea (NRF) grant funded by the Korean government (MSIT) (No. 2020R1F1A1065626) and was partly supported by the MSIT (Ministry of Science and ICT), Korea, under the ITRC (Information Technology Research Center) support program (IITP-2020-2018-0-01798) supervised by the IITP (Institute for Information & communications Technology Promotion). It was also partly supported by the research fund from the Biomedical Institute for Convergence (BICS), Sungkyunkwan University.

**Institutional Review Board Statement:** The study was conducted according to the guidelines of the Declaration of Helsinki, and approved by the Institutional Review Board of Kangbuk Samsung Hospital (No. KBSMC 2019-01-014).

**Informed Consent Statement:** Informed consent was obtained from all subjects involved in the study.

**Data Availability Statement:** The ISBI2018 data presented in this study are openly available in MDPI at https://doi.org/10.3390/data3030025 (accessed on 19 March 2021), reference number 14. Publicly available datasets were analyzed in this study. The EyePACS data can be found here: https://www.kaggle.com/c/diabetic-retinopathy-detection/data (accessed on 19 March 2021).

**Conflicts of Interest:** The authors declare no conflict of interest.

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
