# Peer review of "Future Image Synthesis for Diabetic Retinopathy Based on the Lesion Occurrence Probability"

_electronics, doi:10.3390/electronics10060726_

Round 1

Reviewer 1 Report

The authors proposed to synthesize future fundus image by considering the progression of the disease with two steps training manner. They obtained a better results compared with previous methods.

  1. Why this was submitted to Electronics? Other journals such as medical imaging or diagnosis may be fit to the content better.
  2. Is the prediction of future image so important? Prognosis may be more important than image.

Author Response

Please refer our response of Reviewer #1 in the attached file of our response letter.

Reviewer 2 Report

This is a very interesting article describing the use of deep learning to produce a prognostic image for patients with diabetic retinopathy. The authors used several different architectures to synthesize future retinal images and to make a prognosis for future disease severity. 

The study is in an up-and-coming field, with a relatively small number of papers in it in the recent years. There is a very recent paper by Arcadu et al (https://www.nature.com/articles/s41746-019-0172-3) that the authors may wish to discuss, but it is certainly a field open for innovation. 

The paper seems to have a solid methodology. That being said, I found it very difficult to read. Other than the obvious need for English editing, several issues were unclear to me. The very first one was the availability and use (or not) of longitudinal data. It is unclear to me what the "future fundus" images are, and what the input used to produce them is. The authors do make a good point of registration problems in longitudinal datasets, and therefore for the need to use extracted models, but in the end, it is unclear to me what exactly they end up using. 

Results are presented qualitatively in figures 2 and 3 - as a comment here, I found the images too small to be able to see something on them, maybe presenting fewer cases in more detail would be helpful. Quantitative results are presented in Tables 1-5, and, while it seems like their technique outperforms similar ones in literature, I had trouble figuring out what data they used to produce their results in the comparison. 

Overall, I think this is a very promising study, however the way the paper is structured and poor English makes the paper very difficult to read for me. Inserting some more detail to make clearer what exactly data were used and the exact tests that were run would help me enormously. Also, some better figures where you can see the individual images at a reasonable size would be helpful. 

Author Response

Please refer our response of Reviewer #2 in the attached file of our response letter.

Reviewer 3 Report

The authors present an exciting research with good results on predicting the occurrence of DME in advance. The use of GAN networks for synthetic fundus image generation and their subsequent use for generating future fundus images are definitely thought-provoking. However, many details must be improved in order to validate and understand the proposed approach. These are listed below:

  1. Lot of minor grammatical errors throughout the text. Improve English style and avoid long confusing sentences.
  2. Significant lack of relevant references. Such as GAN networks used similarly or otherwise on other medical image applications.
  3. Very little information on what and how the interval vector is calculated. As it is an important step towards lesion probability predictor, it needs to be explained better.
  4. For Figure 1, it might be also good to visually show how the update of the models are performed. (This figure is otherwise very good.)
  5. On line 108, “the lesion mask indicates four lesions”. Please explain this!
  6. All equations need further explanation and references. Many terms in the equations are not immediately explained.
  7. An appendix/table with all acronyms explained, will be good.
  8. Dataset explanation is confusing. Give total and final numbers on how much data was used for different models’ training, validation and testing.
  9. In ISBI2018 dataset: have you only used disease grading data? This needs to be made clear.
  10. D1 was trained for 100 epochs and D2 was trained for 2000 epochs. Give details on why this was chosen.
  11. Give a brief explanation of the previously developed vessel segmentation model in this article.
  12. For qualitative results, it would make better sense to have a clinical expert look at predicted images to see if they look similar to original images, with a qualitative rating.
  13. It would be nice to have a discussion section at the end of results to explain your understandings from the various results. Also, lot of result explanations written elsewhere in the paper can be brought together under this section.
  14. Important to show best and worst prediction images and give discussion on it.
  15. The authors have used 3 DR severity classifications, but the datasets have 5 classes of DR severity. Why is there a difference?

Author Response

Please refer our response of Reviewer #3 in the attached file of our response letter.

Round 2

Reviewer 3 Report

I appreciate the authors in making all the extensive changes as suggested and commented by the reviewers. I believe the paper is now ready to be published. Good luck!